# Urban community health workers in Punjab, India: A qualitative study of ASHAs' roles in the health system

Baldeep K. Dhaliwal[1,2]*, Madhu Gupta[3], Anuradha Nadda[4], Shalini Singh[1], Anita Shet[1,2], Kerry Scott[1,5], Aakanksha Dutta[6], Svea Closser[1]

**1** Department of International Health, Johns Hopkins Bloomberg School of Public Health, Baltimore, Maryland, United States of America, **2** International Vaccine Access Center, Johns Hopkins Bloomberg School of Public Health, Baltimore, Maryland, United States of America, **3** Department of Community Medicine and School of Public Health, Post Graduate Institute of Medical Education and Research, Chandigarh, India, **4** Department of Community Medicine, Dr.BR Ambedkar State Institute of Medical Sciences, Mohali, Punjab, India, **5** School of Global Health, York University, North York, Ontario, Canada, **6** Centre for Public Health, Panjab University, Chandigarh, India

* bdhaliw1@jhu.edu

## Abstract

The Accredited Social Health Activist (ASHA) program is the world's largest all-female community health worker (CHW) initiative. While most CHW programs have been extensively studied in rural contexts, little is known about how ASHAs and CHWs operate in urban settings. Research on urban programs globally remains limited with a primary focus on single-disease interventions. A more holistic understanding of urban ASHAs' roles is needed to more comprehensively understand urban health delivery. This study explores the experiences, challenges, support systems, and systemic barriers faced by urban ASHAs in Punjab, India. This qualitative study was conducted in one urban and one peri-urban site. Data collection included 25 in-depth interviews, participant observation with 28 ASHAs over three months, and community-level focus group discussions. Data were analyzed using thematic coding with MAXQDA software. A half-day financial participatory session was implemented to document the financial aspects of urban ASHAs' work. This study documented that urban ASHAs play a vital role in connecting vulnerable populations to healthcare and promoting government health services. Despite this, they face challenges including overseeing populations that far exceed the limits set by guidelines, limited training opportunities, low community engagement, and insufficient financial compensation. Systemic barriers, such as unfilled supervisory positions and minimal collaboration with community engagement structures exacerbate these issues. To maximize the impact of the urban ASHA program, policy makers and implementers may consider strengthening governance, refining ASHA selection processes, enhancing community engagement, addressing staff shortages, providing targeted training, and revising financial incentives. Implementing these recommendations may strengthen urban

**Data availability statement:** The analytic code-book and specific excerpts of the transcripts are available. Data requests may be sent to the BSPH IRB office (BSPH.irboffice@jhu.edu).

**Funding:** This work was supported by the Fulbright-Nehru Doctoral Research Fellowship, a program of the U.S. Department of State, United States-India Educational Foundation, and the Institute of International Education (G2023//ST/14 to BD).

**Competing interests:** The authors have declared that no competing interests exist.

ASHAs' ability to deliver equitable healthcare in Punjab and provide a model for improving urban health delivery across India.

## Introduction

Over the past several decades, the Government of India has committed to improving public health, enhancing health indicators, and responding to emerging health challenges through transformative policy changes. A key component of these efforts was the launch of the Accredited Social Health Activist (ASHA) program in 2005 [1]. Over one million ASHAs, a cadre of community health workers (CHWs), operate across 27 of India's 28 states, forming the world's largest all-female community health workforce [2].

The ASHA program was initially designed to support marginalized rural areas, as these populations face significant barriers to care [1]. Over the last two decades, the rural ASHA program has made substantial strides, with access to ASHA services being associated with a 17% increase in first antenatal care visits, a 5% increase in four or more antenatal visits, a 26% increase in having a skilled birth attendant at delivery, and a 28% increase in facility-based deliveries [3]. Despite these impressive strides in rural India, marginalized populations in urban India faced distinct healthcare access challenges, often without comparable government intervention. To address these growing disparities, the Indian Government launched the National Urban Health Mission (NUHM) in 2013, expanding the ASHA program to support urban populations [4,5]. Despite the urban ASHA program being active for over a decade, research on this program is limited. This mirrors the global literature gap on CHWs' contributions to urban health systems [6].

Globally, most research on CHW program implementation and effectiveness has focused on rural programs [7], despite the rapid urbanization of many countries [8]. Studies on urban CHWs have largely examined their role in specific health issues [9,10] or evaluated their effectiveness in delivering single-disease interventions [11,12]. However, there is little evidence on the broader design, management, and administration of urban CHW programs, including payment structures, supervision, and recruitment policies [9]. Although the urban poor can be as vulnerable to health risks as the rural poor [13], identifying solutions to address growing urban health disparities has not been extensively documented in the global health literature [14]. This gap is particularly pronounced in the ASHA program; while there has been extensive research examining the functioning of rural ASHAs, little is known about how the program operates in urban settings.

The existing literature on urban ASHAs has primarily focused on specific health campaigns, such as eye health education and at-home breast exams [15,16], or more recently, on evaluating their effectiveness in achieving health outcomes based on select performance metrics [17]. A systematic review of 122 studies published between 2005 and 2016 on India's ASHA program found that only two focused on urban settings [18]. As India's urban population grows, understanding how ASHAs navigate healthcare delivery for marginalized populations in these settings is crucial.

Without a deeper understanding of how urban CHWs, including ASHAs, function within complex health systems, efforts to support urban healthcare delivery risk being misaligned with the realities on the ground.

In this study we examined how urban ASHAs' roles, challenges, and support structures align – or fail to align – with the evolving health needs of urban communities in two sites in urban Punjab, India. This was guided by the CHW-health systems interface framework which explores CHW social profile and agencies, CHW program inputs, CHW-community interface, the health services context, program governance, program outcomes, and program impact [18]. This specific framework was selected as we aimed to understand the urban ASHA program on a broader level, which required a detailed understanding of elements of the frameworks' broad domains. We found that while ASHAs are expected to support the most vulnerable populations, inadequate support, socioeconomically diverse populations, limited compensation for transportation, and difficulties with community engagement limit ASHAs' ability to ensure that they are most effectively supporting vulnerable populations.

By developing a more nuanced understanding of the urban ASHA program, this work aims to provide an essential window to begin to understand these broad, complex issues of health delivery among marginalized urban populations, both in India and on a global setting. This research is an essential initial contribution to strengthening global urban community health delivery by focusing on CHW program design, management, and sustainability in India's ASHA program.

## Background: The ASHA program

Launched under the National Rural Health Mission in 2005 – and under the National Urban Health Mission in 2013 – the ASHA program broadly aims to improve equitable access to healthcare and strengthen health delivery systems for marginalized populations in rural and urban areas. As of 2021, a total of 68,931 ASHAs have been selected [19]. The urban mission was designed to improve the health status of urban populations by providing care through a network of Urban Primary Health Centers, Urban Community Health Centers, and urban ASHAs [4,5]. Grounded in the belief that community-based workers drive health behavior change, ASHAs were implemented to leverage local bonds to enhance outreach [20].

All ASHAs are expected to receive at least 23 days of training on basic health topics (i.e., identifying basic health issues, linking members of the community to health services, and mobilizing the community on public health issues including water, sanitation, and nutrition issues) before she is assigned a population of approximately 1000 people to provide care to [1,21]. The training for urban ASHAs encompasses these topics but also requires additional training on health vulnerability assessments, household mapping, and health resource mapping [5]. Urban ASHAs are also trained on key competencies tailored to urban settings, including adolescent health, issues of drug and alcohol abuse, and violence against women. ASHAs in both rural and urban settings are responsible for raising awareness about health-related social determinants, such as nutrition, sanitation, and hygiene, and providing support to marginalized groups. They also facilitate continuity of care through home visits, facility escorts, and outreach activities like Health and Nutrition Days. ASHAs are also expected to facilitate coordination with community-based organizations, such as Mahila Arogya Samitis (MAS) – or women's collectives in urban settings – to strengthen community ownership of health interventions and empower women as active health advocates [4,5]. Although ASHAs receive a modest honorarium which varies by state, most of their earnings come from task-based incentives tied to specific maternal and child health services.

The ASHA program is set up in hierarchical fashion (Fig 1).

ASHAs primarily report to ANM (Auxiliary Nurse Midwife) workers, who complete a two-year certification program for this role. Positioned as intermediaries between ASHAs and higher-level health officials, ANMs oversee ASHA activities, ensure task completion, and provide guidance on maternal and child health initiatives.

## Methods

This qualitative study draws on several data sources: (i) in-depth interviews (IDIs) and community-level focus group discussions (FGDs), (ii) participant observation, (iii) case studies, and (iv) a finance participatory session. This work was

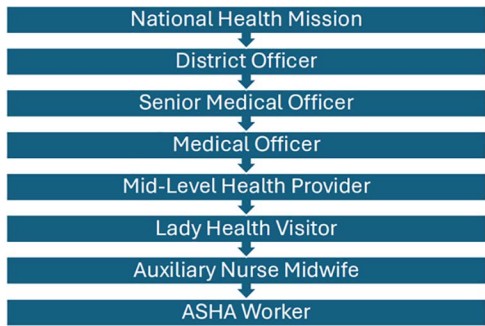

**Fig 1. Hierarchy of State Health System.**

conducted by the first author (referred to as "BD" and "I") between September-December 2023, with rapport building in late September and initial participant recruitment starting on 04/10/2023 and ending on 07/12/2023.

### Ethical considerations

The research team obtained approvals from the Johns Hopkins Bloomberg School of Public Health (BSPH) (IRB #25369), Panjab University (ECR-2308–162), and the Punjab National Health Mission (NMH/PB/CCP/2023/106613–16). BD took written consent from all participants, explaining the purpose of the research, the voluntary nature, and any risks and benefits using an informed consent form and participant information sheet in Punjabi or English. Moreover, in accordance with the American Anthropological Association's Statement on Ethics and Principles of Professional Responsibility, BD emphasized ongoing informed consent as a part of the study design to promote participant agency and engagement [22]. As ethnography has the potential to blur lines between research and friendship [23], BD engaged in ongoing discussions to clarify whether participants intended their shared experiences to be treated as research data or as personal exchanges; data not considered as research by participants was excluded from field notes. After receiving ethical approval, we reported one protocol deviation on November 10, 2023, when a group interview was conducted instead of an individual interview. The BSPH IRB determined that this was a minor non-compliance that did not affect participants' rights or welfare.

### Field sites

The study was conducted in two sites in Punjab, India which were selected by the National Health Mission, Punjab. The site referred to as the "urban site" in this text was an urban city of roughly one million people that has seen rapid development in recent years. The area is characterized by a growing number of shopping malls, business centers, and newly constructed residential apartment complexes, reflecting its transition to an urban hub. This city is undergoing significant economic and infrastructural expansion, attracting a population of wealthy individuals seeking to relocate, as well as the migrant workers who are facilitating this transition. Migrants often lived in sprawling slum settlements where homes were pieced together from available materials – discarded construction debris, tarps, and rusting sheets of metal. Some of these settlements have stood for years, while others emerge overnight near construction sites, quickly assembled as temporary shelters for migrant workers. The roads within these settings are typically uneven dirt paths, at times pooling with wastewater. Access to water typically comes from a shared hand pump, where women gather daily to fill containers. With limited sanitation infrastructure, open defecation is common. Income is unstable with most women finding work as domestic helpers in nearby homes, while men take on day labor in construction, drive rickshaws, or engage in other forms of informal, low-wage work. The site referred to as the "peri-urban site" in this paper was an area with a population of approximately 500,000 people. The peri-urban site is characterized by its unique blend of rural and emerging urban elements, attracting a significant number of migrants for agricultural work and employment in local industrial positions. While

some neighborhoods resemble organized areas with concrete homes and small shops, others resemble the slums of the urban site. Both areas have a complex socio-economic environment, where long-term residents and new arrivals navigate shifting job opportunities, infrastructure challenges, and access to basic services. We have chosen not to name the study sites in order to preserve participant anonymity and safeguard the identities of the ASHAs involved in this research.

### Data collection

The research team and public health experts introduced BD to ASHA supervisors in-person across both field sites, who in turn facilitated introductions to ASHAs. Following these initial meetings, BD began study activities, meeting ASHAs at their convenience and accompanying them in their daily routines. BD conducted participant observation with 28 ASHA workers across the two field sites, spending several hours a day, six days a week, shadowing their routine activities such as door-to-door community outreach, vaccination sessions, hospital visits, survey work, ad-hoc tasks, meetings, and interactions with community members and other health workers. As an "active observer," BD participated in ASHA activities, conducted informal and formal interviews during this time, and built rapport through social engagement.

### In-depth interviews

BD conducted 25 formal in-depth interviews until thematic saturation was reached: 13 with ASHAs and 12 with policymakers and health program administrators. All participants were initially recruited through the research team and local health system contacts, followed by snowball sampling to identify additional stakeholders; participants were purposively selected based on their knowledge of the ASHA program. Interviews with ASHAs largely explored their motivations for remaining in their roles, their expected responsibilities and tasks assigned to them, their interactions within the community and health system, and alternative opportunities available to them. Interviews with policymakers and administrators focused on program design, management, and administration, and included state and district health officers, medical officers, ANM workers, and national-level health officials. These stakeholder were selected based on their relevance to the ASHA program's design and implementation. Interviews ranged from twenty minutes to two hours. I continued to facilitate recruitment and conduct interviews until I reached thematic saturation, or no additional data were found [24]. Since the concept of 'saturation' in qualitative research lacks a structured framework, I assessed saturation using 'conceptual depth criteria' to provide a systematic approach [25]. In line with these criteria, I determined that saturation was reached when the data contained multiple instances illustrating key insights, demonstrated subtlety to convey deeper meaning, and exhibited validity through confirmation of results from individuals within the context. Most interviews were conducted by BD in Punjabi, with two stakeholder interviews in English. Interviews were audio recorded; two stakeholders declined to be recorded, and detailed field notes were taken for those interviews instead.

### Financial data collection exercise

We conducted a half-day participatory session in the urban site to document the financial aspects of urban ASHAs' work. Twenty-five ASHAs attended the session and completed a hand-written spreadsheet to record the time they spent on each work-related task, the financial costs they incur when completing tasks, and the incentive amounts they earn for completing each task. ASHAs were monetarily compensated for their participation, both in recognition of their time and to offset any potential loss of incentives.

### Data management and analysis

While in the field, I took jottings during interactions subtly, which were later expanded into detailed field notes on a regular basis. These notes captured reflections on daily activities and helped identify emerging questions for further exploration

[26]. I transcribed and translated all interview recordings. Each week, I systematically revisited my research questions and study aims and conducted a rapid analysis of field notes and transcripts to assess whether key aspects had been sufficiently explored or if gaps remained. Based on these reflections, I adapted my data collection approach by refining my data collection tools – adding new questions to probe emerging themes, modifying prompts to ensure richer responses, and removing questions that were ineffective. This iterative process ensured that the research remained dynamic and responsive, allowing me to delve deeper into areas. After completing data collection, I began thematic analysis with a close reading of the data and development of a coding framework (list of codes with their definitions, grouped by topic) [24,27]. Coding followed a hybrid approach, drawing deductively from the CHW–health systems interface framework [18], which was developed primarily based on literature and programmatic experience from rural settings. To account for new findings, as well as the distinct features of the urban ASHA context, additional inductive codes were developed to capture emergent themes. After developing and refining the coding framework, I applied it to all the transcripts using the qualitative data management software MAXQDA [28]. I then read coded outputs to identify higher-level themes, such as "ASHA Payment" or "ASHA Community Benefits", with subcodes that fell into these categories. These themes were developed and substantiated with descriptive quotes from the data to form a codebook. This codebook was applied to all transcripts and field notes. I also analyzed the completed spreadsheets from the finance session to calculate averages for time, costs, and incentives across groups, providing a descriptive understanding of ASHAs' financial experiences.

## Results

This section provides a structured account of the urban ASHA experience across two study sites, addressing knowledge gaps in how the program functions on the ground. We offer a descriptive overview of key aspects of the ASHA role – capturing the complexities of their daily work and the structural factors that shape their effectiveness. Our findings begin with a short case study that highlights day-to-day realities of urban ASHAs. We then discuss experiences during the ASHA selection process, health system support structures, initial training processes, the community context and engagement within the community, their financial incentives, and perceived benefits of ASHA work.

### Sabita's story

One sunny morning, Sabita (*pseudonym*), an ASHA, and I met at 10am in the urban site. At 29 years old, she had been an ASHA for three years – as a wife and a mother of two young children, she was eager to support her family as her husband was recovering from medical issues. We started our morning cautiously navigating *kothis* (mansions) of wealthy residents to make progress on her population survey – something she is expected to complete monthly for everyone in her area, regardless of their financial background. She carefully documented the number of people in each household – noting if any of them suffered from a non-communicable disease – and checked for any unvaccinated children in the house.

"*I'm still nervous to go to kothis alone,*" she said as we walked. "*The people are educated and get mad at me for coming because they think I'm talking to them about government services. They say they don't need them.*"

Many doors were shut in our faces, and several people shouted at us, refusing to share their '*biodata*'. Frustrated, Sabita decided to switch to her maternal and child health (MCH) tasks. "*This survey work doesn't even give me an incentive,*" she said as we waited for a rickshaw. "*At least my MCH work does.*"

Ten minutes – and 40 rupees of rickshaw fees later – we were in a more modest neighborhood. We navigated the streets to a smaller home where a family rented the first floor of a three-story house. The father-in-law welcomed us in, recognizing Sabita. He led us to a bedroom where a woman lay resting with a newborn baby nearby. Her questions came out rapidly, checking on both the mother and child, "*Has she been feeding? You're going to bring her for vaccines in two weeks, right? Come on Tuesday to the clinic and bring her card. Are you having any pain after delivery?*"

After ensuring everything was in order, Sabita took a quick picture with the mother and baby – evidence for her supervisor that she had completed this task – and we prepared to leave. Her father-in-law stood nearby with a plate of

pink '*ladoo*', a treat that often signifies the birth of a new baby; he smiled and insisted we take one to welcome his new granddaughter.

With our mouths still sweet from the *ladoo*, we took another rickshaw to a small settlement which also fell in her area. She was most at ease in this settlement, considering her own family lived nearby in a similar – but slightly more established – setting. In just 20 minutes we (1) sought out and checked on a woman who was badly injured in a road accident; (2) conducted a wellness check on a woman who had suffered a miscarriage; (3) guided a woman on how to obtain an *Aadhaar* (identification and social security) card after hers was lost to flooding; (4) made a plan to take a woman to the district hospital to get a new birth certificate for her child; (5) completed a postnatal visit for a newborn; and – while walking between these tasks – (6) encouraged mothers of small children to attend the upcoming immunization session. Although these tasks clearly bridged community members to the health system, only one of these tasks provided Sabita with a financial incentive.

As Sabita and I carefully stepped around '*nallahs*' –small streams of rainwater, wastewater, and sewage – and swatted away mosquitoes, I reflected on the words of a senior ASHA supervisor who had shared during an interview, "*The people in slums are totally dependent on the ASHAs. They see them as doctors because they are the <u>only</u> ones who can help them*."

By the time we reached the health clinic at 4 PM, Sabita's day had already spanned three neighborhoods, multiple health-related tasks, and countless interactions with community members – yet much of her work remained invisible and unpaid. From navigating the *kothis* of the wealthy, where her presence was met with resistance, to the informal settlements, where she was the first point of contact for healthcare and social support, her role adapted to the needs of each setting. While she was tasked with bridging vulnerable populations to essential health services, the burden of unpaid tasks, out-of-pocket costs for transportation, and community resistance in wealthier areas highlight the challenges ASHAs face in fulfilling their mandate.

While ASHAs are expected to support the most vulnerable populations, inadequate support, socioeconomically diverse populations, limited compensation for transportation, and difficulties with community engagement limit ASHAs' ability to meet the mandate put on them by the health system.

### "I thought why don't I just put in an application?": The ASHA selection process

Like other CHWs globally, ASHAs are intended to be selected by and from the communities they serve, ensuring stronger community ties and engagement. This participatory selection process is designed to allow ASHAs to be recognized as trusted intermediaries, fostering deeper relationships with local families and improving health outcomes. However, participants in both sites reported that the ASHA selection process lacked community-driven initiatives.

When asked about recruitment, ASHAs consistently mentioned a connection who informed them of a vacancy and encouraged them to apply. An ASHA in the urban site, Bhavna, explained how she just applied to her role. *"I used to go to the doctor here [at the dispensary] to get medicine,"* Bhavna explained during an interview. *"When I told the doctor that I do stitching, I started working for them a bit. Through these dispensary visits [dropping off and picking up stitching], I got to know the lady who worked here a little. She came to my house one time and said, 'we need some ASHAs for this area if you're interested'. I thought why don't I just put in an application?"* Bhavna started her role as an ASHA soon after, and she had been working for the past four months.

ASHAs in the urban site were initially recruited mid-2020 to address pressing COVID-19 needs. While the urgent need for ASHAs explains the lack of community engagement in their selection processes, ASHAs in the peri-urban site – who began in 2016 – also reported finding their roles in a similar way.

*"When my father-in-law was sick, he was in the hospital for about a month,"* an ASHA of seven years told me as we navigated her area in the peri-urban site. *"I would go every day and help take care of him, and then I also took care of*

*the other patients as well. While I was there, the hospital said, 'you know, we're starting to look for urban ASHAs in the area that you're from, would you be interested in being an ASHA?'."*

She laughed at the memory. *"I didn't know what an ASHA was then. I said, 'I don't know. What is an ASHA? What do they do?'. They said 'it's basically the same thing that you've been doing. You take care of sick people, and you help them, you bring them to the hospital. You're in the community, people already know you, and you're already here helping people. You may as well do it for money'."*

Although this hiring process may have been more straightforward, it may have reinforced the perception of ASHAs as extensions of the health system rather than embedded community representatives.

## Health system level support for ASHAs

ASHAs rely on system-level support to effectively carry out their work. The health guidelines outline a structured framework of supervision and mentorship with Medical Officers, Auxiliary Nurse Midwives (ANMs), and Lady Health Visitors (LHV) providing oversight, training, and problem-solving support. Although this framework should ensure that ASHAs can navigate their responsibilities and challenges, across both urban sites ASHAs reported that these structures offered varying levels of support.

A newer ASHA, Ananya – who had been in her role for six months – painstakingly noted which households refused to answer her survey questions and took their signature as evidence that they didn't want to answer. *"When I leave the page blank, I get in trouble,"* Ananya explained. *"Even when I tell the Medical Officer that people refuse to give me their information, sometimes she still goes and checks with them to see if that's true."*

In both sites, the absence of a designated city-level community mobilizer further exacerbated these challenges. This position, intended to provide targeted community engagement support, remained unfilled, creating a critical gap in the chain of supervision and mentorship. As a result, much of the day-to-day management of ASHAs fell to the ANM workers – who often acted as a direct supervisor for ASHAs.

ANM workers reported being systematically overburdened, with Rani – an ANM worker of over five years – reporting that there were only three working in an area with 150,000 people. *"According to the guidelines, there should be 15 ANMs,"* she told me. Rani was frustrated by this asking, *"How can we be expected to give high quality care this way?"* Despite this burden of care, and their limited time, ANM workers like Rani had taken on mentoring and supporting ASHAs.

When Anjali – an ASHA worker of three years – was detailing her frustrations with the health system, she emphasized the support Rani offered her. *"A few days ago someone working at the hospital was saying to me 'you don't do work, so you don't need to come to the hospital'. Then I went to Rani Mam crying, and she said 'no child, it's not like that. Just keep working hard, and people will stop [saying these things]'. Our ANM mam is so supportive. If I have any issues, the seniors listen and help me,"* highlighting the supportive role of ANM workers and others in the health system.

Further, ANMs were generally beloved by the ASHAs, constantly encouraging and guiding them despite the overwhelming workload. During an interview, an ASHA at the peri-urban site explained how much they appreciated the ANM worker saying, *"This ANM who you just met is our mam – but we call her didi (big sister)!"*

Despite the limitations of formal support structures, ASHAs built their own networks of solidarity, relying on trusted ANM workers and each other for guidance, encouragement, and problem-solving within the broader health system.

## "When a new ASHA joins, we teach each other": ASHA training procedures

ASHAs are expected to undergo structured training to equip them with the knowledge and skills needed to support maternal and child health, conduct community outreach, and facilitate access to health services. However, ASHAs across both sites reported that formal training was often delayed – or in some cases, never conducted – leaving them to rely on peer learning and on-the-job mentorship to navigate their roles.

One ASHA who started in her role a year prior, explained that the expectation is that they support each other, "*Well, didi (big sister) taught me this because we don't have a formal training session. When a new ASHA joins, we teach each other [like didi taught me]. The ANM Ma'am helps at first, but we mostly rely on each other.*"

ASHAs reported that their ANM provided initial practical training, guiding them through their assigned areas, explaining their responsibilities, demonstrating how to communicate with pregnant and new mothers, and instructing them on how to approach households for survey work. An ASHA explained in an interview, "*They [ANM workers] explain everything and go with us over 2-3 days, and then the other ASHAs train us. Then we can always go back and ask more questions.*"

An ANM worker of ten years elaborated on this, explaining, "*A formal training is later after starting because we expect the ASHAs to learn in the field. There is an expectation that they pick things up and teach each other.*"

A district officer confirmed that there were occasionally delays in the training process, so this ASHA-led training was used in the meantime: "*When any new ASHA starts, they all get an eight-day training. But it's true it might not happen right away. So, in a month or two we might have 2-3 ASHAs join. We can't train them all as one-offs. We have to make a batch with a minimum of 40 ASHAs, and they should get this eight-day induction training – maybe not immediately but it happens within the year.*"

This occasionally did not happen, with one ASHA who had been in her role at the urban site for three years, reporting, "*Fine. I understand that I started during COVID and there weren't staff. There weren't people who were offering the training. But there's still nothing now.*"

Despite the absence of formal onboarding training for ASHAs, there were consistent training sessions for new tasks and responsibilities. One ASHA reported, "*I had one day of training with the ANM worker. In this session they went over everything at a basic level like area and responsibilities. Other than this, I've only had trainings for schemes (programs) or when issues – like checking dengue hotspots – come up.*" A LHV confirmed this explaining, "*As work comes up, we do training. When we get instructions from the hospital, we bring them all to the district hospital and have their training there.*"

Through informal peer-led learning networks, ASHAs ensured that essential training was passed down through experience and collective support, even in areas where structured instruction was lacking.

**"How can one single ASHA manage?": The community context**

The urban ASHA program was designed to focus on the most vulnerable populations, ensuring that those with the greatest health needs receive dedicated support. In practice, however, ASHAs found themselves responsible for entire neighborhoods, often without clear guidelines on how to differentiate or prioritize vulnerabilities among residents. Administrative demands, such as mandatory survey work and additional assignments from health officials, further expanded their scope, requiring them to engage with all households – regardless of need.

As I sat in an office interviewing an ANM worker about the design of the ASHA program, a frazzled ASHA ran into her office, filled her bag with folic acid packets for pregnant mothers, and darted back out. "*That ASHA who just came in,*" the ANM started. "*Her entire population is 13,000. How can one single ASHA manage? The workload is so high they can't do good work.*" However, these 13,000 people were not all vulnerable and varied widely from a socio-economic perspective. Sabita's story was not an isolated incident, and most ASHAs navigated complex socio-economic landscapes, balancing their intended role as community health connectors with a growing burden of administrative and outreach tasks that extended far beyond the most vulnerable populations. ASHAs were required to visit every household, including affluent families – who neither rely on nor welcome government healthcare services – to complete survey work, while still being expected to support vulnerable populations in need of health assistance.

This varying socio-economic context among community members was also common in the peri-urban site. During participant observation with Aditi, an ASHA of eight years, we walked through a temporary settlement. Aditi would call out to women who had recently moved to the settlement, "*Hello di (sister)! Is that your child? How old are they? Do they have a vaccine card?*" None of them did. "*Bring them to the vaccine clinic at the health center today. We'll help you make sure your child is caught up.*" The mothers usually agreed.

The setting and population rapidly changed as we walked. The homes began to look more established, each with multiple rooms, courtyards, and fenced in areas where they parked their motorbikes or, for some, their cars. Seeing my confusion, Aditi explained. *"These people have been here for 10-15 years now. They've been able to send their kids out. A lot of their kids are now in the army, and they have good jobs. Some even have government jobs! They're all doing well, and they've built themselves up from what they used to be. But when they first came, they were like the people on the other side."*

In this more developed area, Aditi's visits took on a different tone. Residents often invited us in for tea and snacks, warmly welcoming her survey questions, even though they no longer required her MCH services. Many women had already consulted private doctors for their pregnancies and taken their children to private doctors for vaccines. However, unlike in the urban site – no doors were shut in our faces, and no one was offended by the idea of government services. While these interactions showcased Aditi's valued community presence, like the ASHAs in the urban site, she was spending time supporting populations that were no longer the most vulnerable.

The role of ASHAs in the community raises a critical tension in the vision of primary healthcare. Ideally, government health services should be accessible and utilized by all, fostering trust and engagement across socio-economic groups. In reality, ASHAs are often compelled to focus their health-related efforts on the most vulnerable and forced to target wealthier populations with administrative tasks with wealthier populations. This dynamic not only limits ASHAs' ability to navigate wealthier communities but also perpetuates the notion that public healthcare is inferior and designed for the poor. While prioritizing the most underserved populations is a practical necessity in a resource-limited system, it also risks deepening existing inequalities in healthcare access and perception.

### *"They felt they didn't get anything out of it"*: Difficulties with *Community Engagement*

ASHAs play a crucial role in bridging the gap between communities and the health system, fostering trust and encouraging participation in government health programs. Although ASHAs are supposed to be supported through coordination with local leaders and structured platforms, in practice, ASHAs struggled to gain community trust and engagement, facing limited coordination from community leaders, skepticism from residents, and declining participation in formal community meetings.

The urban site was split into different neighborhoods, each with a distinct community council referred to as a Municipal Corporation (MC). Each council was led by an elected man or woman, a person who was simply referred to as '*the MC*'. While the MC was known and respected within the community, most ASHAs reported that these leaders were not actively involved in supporting their work or integration into the community. As I walked through an area with Sapna, an ASHA of four months, she was frustrated by this limited interaction. *"The MC should tell people the ASHAs are coming. He should tell them our names so they know – this way they would have no problem with us coming,"* she frustratedly said as we'd had yet another door shut on our face. *"The MC has the authority to do it – but no one has done this. Sometimes people say to us 'the MC has not informed us you are coming', so people don't want to give their details. Then we have problems."*

However, in the more vulnerable parts of an ASHAs area, community members felt that the ASHA was their key connection to healthcare and the health system. *"She is the only one who comes and checks on all of us, checks our kids are okay, checks that we're okay,"* a FGD participant from a slum reported. *"It doesn't matter if someone has older or younger kids, she asks about everyone. No one has health issues with her here."*

Across slums and peri-urban sites ASHAs are expected to facilitate community engagement through the MAS; however, the MAS was often simply used for community education, as opposed to a community-driven engagement platform. An interview with a district-level stakeholder confirmed this. *"In the village they have Village Health Sanitation and Nutrition Days. These [MAS meetings] basically have the same function. They do the same things, but the name is different."*

An older ASHA in the peri-urban site explained that while there used to be engagement with the MAS, it eventually stopped. "*Over time, people stopped showing up to the MAS meetings,*" she said. "*There was frustration because they felt they didn't get anything out of it.*" Among newer ASHAs, most had no idea about MAS meetings, gently responding, "*I don't think we have this in my area,*" when asked.

Without strong institutional backing or meaningful community incentives, ASHAs struggled to engage residents, leaving many community platforms underutilized and reinforcing their position as peripheral rather than central actors in local health governance.

### Payments and incentives

Although ASHAs play a critical role in delivering community healthcare, their compensation remains low, unpredictable, and often insufficient to cover the costs of their work. ASHAs across both sites reported that they received a guaranteed honorarium of 2500 rupees per month (less than $30 USD), and they earned incentives for additional tasks. However, as transportation costs associated with traveling to complete their tasks are not reimbursed, ASHA workers explained that this honorarium – something they often referred to as their 'salary' – typically only covered transportation expenses.

One ASHA reported, "*The [guaranteed payment] is almost fully lost in the coming and going. The late-night rickshaws I take when a community member goes into labor in the middle of the night, the back and forth to hospitals with pregnant women for visits and going between areas.*"

Another ASHA expressed her frustration as we walked towards a house to drop off a maternal immunization card for an expectant mother. "*We only get one incentive, but how many rounds does it take? I went there to get the card made – 20 rupees there, 20 rupees back – and that's one of my many visits for her, all to get a 150-rupee incentive [when she] delivers the baby.*"

In addition to the maternal and child health work outlined in their guidelines, ASHAs are regularly assigned ad-hoc work, such as outbreak investigation, registration for government programs, and survey work. Occasionally, ASHAs reported being promised incentives for ad-hoc work – such as election work – that did not materialize. One ASHA shared her experiences saying, "*The last election we worked from 7am-7pm because they told us they would give us 500 rupees to work all day. We told them that we needed it up front because they never pay us after,*" she said, her words coming out faster the angrier she became. "*They told us they would give it to us at the end of the day – then you know what happened? They left right after the election ended. They told us it should be on in our pay this month.*" Her tone changed, and she sounded defeated. "*Will it happen? Who knows?*"

During the financial participatory session ASHAs provided an in-depth look at many of the tasks they are responsible for (Table 1). In addition to their base honorarium, ASHAs earn incentives for select tasks; however, ASHAs reported that the time dedicated to these tasks varied significantly, from under an hour for certain postnatal care visits to nearly ten hours for overnight stays following a child delivery. Further, ASHAs reported a sharp disconnect between the costs incurred to complete the task and the incentives provided. For instance, antenatal care visits consistently resulted in a financial loss, as the expenses associated with transportation and other necessary resources outweighed the modest incentives. Only a few activities, such as child delivery, overnight stays after delivery, certain vaccines (1-month and 9-month doses), and specific well-child visits (6-month), generate a positive net profit. Even tasks that yielded a profit – such as child deliveries and overnight stays – provided minimal returns considering the time required.

### 'ASHA work brings me out of the dark': Perceived Community, Health System, and Personal Benefits of the ASHA Program

Although urban ASHAs were introduced to the health system relatively recently, their impact extends far beyond their immediate responsibilities. By increasing awareness of government healthcare services and strengthening trust between

**Table 1. Workload and Earnings of ASHAs in Maternal and Child Health.**

| Task Category | Time Commitment | Financial Outcome |
|---|---|---|
| Antenatal Maternal Health (*Four Care Visits*) | 2 + hours per visit | ASHAs report losing between 11 and 56 rupees (*less than $1 USD*) per antenatal care visit after factoring in transportation expenses. |
| Childbirth and Post-Delivery Support | 8 – 10 hours per case | After accounting for travel costs, ASHAs report a net gain of 15 – 22 rupees (*less than $0.50 USD*) for supporting a mother through childbirth. |
| Newborn and Child Health Visits (*Ten Visits per Child*) | Approximately 1 hour per visit | ASHAs visit children at 3, 7, 14, 28, 42 days, as well as 3, 6, 9, 12, and 15 months. Each visit was reported to result in losses of 2.5 – 25 rupees (*less than $0.50 USD*) after accounting for transportation expenses. |
| Immunization Support (*Eight Visits per Child*) | 1.5 – 2 hours per visit | Although ASHAs support caregivers with vaccinating their children for all required doses they report most vaccination visits result in losses of 10 – 63 rupees (*less than $1 USD*) after accounting for travel costs. However, 1-month and 9-months vaccine visits resulted in net benefits for ASHAs, ranging from 28 – 33 rupees (*less than $0.50 USD*). |

communities and the health system, ASHAs have become essential health intermediaries while also finding personal fulfillment and a sense of purpose in their work.

One stakeholder explained how one of the most valuable contributions the ASHAs have made is in educating the public about the benefits of government healthcare in an interview, saying, *"The government puts so much work and effort into launching these good schemes (government programs) – when the ASHA goes door to door then they talk about these schemes. They come here [dispensary] once, and then they won't even try to go to private again. An ASHA will educate them that they can get treatment for TB free for 6 months. If they go private, one month of medicine is 40,000 rupees – a poor family could not afford this! The government has so many schemes – the ASHA explains it, and then they come to the government sector."*

ASHAs also took pride in educating the public about government healthcare options. A newer ASHA in the urban site reported, *"I like to motivate people that they can get everything that they are getting in the private sector in the government for free. They tell me that they went and got these injections for 5000 or 6000 in the private sector, and I like telling them that they can get these injections for free from the government. I like that I am learning, and then in turn I am teaching."*

The health system also benefited greatly from the ASHAs, as stakeholders reported that ASHAs were instrumental in reducing maternal mortality rates (MMR) among migrant populations in the peri-urban site. One ANM worker reported, *"Initially the migrant populations didn't want to come or listen. They never agreed to anything. Now they know the ASHA is there to help them. They even motivate each other! They tell others, 'Oh she [ASHA] is helpful. She took me to the hospital to deliver my baby', and now home deliveries in the slums have gone way down."* This improvement was supported by a district-level stakeholder, who shared, *"Last year our district did commendable work. In the entire country, [our] district had one of the highest reductions in MMR. The ASHAs did a lot for this – they registered women in a timely manner, they sent high-risk pregnancies to the hospital and helped anemic women."*

ASHAs themselves felt that they got mental health benefits from ASHA work. *"At home we have our own tensions – we are just thinking and thinking, and nothing comes of that. Now we go out and we meet so many people – 100 or 200 people a day sometimes! We leave all the stresses of home behind, and we meet new people, learn new things. It's nice."* Many other ASHAs also reported that this job gave them purpose, allowing them to leave their homes, and improve their mental health. *"When you stay home you feel depressed,"* an ASHA told me as we stood in her kitchen carefully watching the *chai* (tea) boil. *"When I go out into the field – I meet the doctors, patients, and I understand the hospital. It's good for me."* An ASHA who had moved from a rural community to the peri-urban site after marriage explained that the isolation and adjustments to motherhood made her lapse into a depression for three years. Since starting her ASHA work, she's 'come out of the dark' by talking to people and not 'sitting at home all day.'

Their relationships with each other and others in the health system also motivated them to stay in their roles. An ASHA responded to my question about relationships eagerly and without needing to be probed. "*Oh, we are like family members,*" she said with a smile. Another told me, "*Yes, we can't live without them, and they can't live without us.*"

The impact of ASHAs extends beyond healthcare delivery – they have become essential links to the health system and sources of support for one another. Despite the challenges of their work, many ASHAs remain motivated by the relationships they build, the lives they improve, and the personal sense of purpose their roles provide.

After a particularly long and difficult day with an ASHA, I asked her why she didn't leave this work when her hours were so long, and her pay was so low. She smiled and her words radiated with the love she feels for the community: "*You've seen how much love people have for us. You can't bring yourself to leave when you see all that love.*"

## Discussion

This study contributes to the growing but still limited global literature on urban CHW programs by offering a nuanced analysis of how urban ASHAs navigate healthcare delivery in complex and under-resourced settings. While rural CHW programs have been extensively studied [7,18], urban CHWs operate in environments characterized by greater socioeconomic diversity, fragmented health systems, and rapidly shifting community needs – underexplored challenges in global health research [9]. Existing studies on urban CHWs have largely focused on the role of CHWs in disease-specific interventions or health education campaigns [9,10], rather than examining the broader structural and governance challenges shaping their work. By addressing gaps in program design, supervision, financial sustainability, and community engagement, this study not only enhances understanding of India's urban ASHA program but also provides critical insights for exploring CHW programs in other urban contexts globally, particularly as countries work toward achieving universal health coverage in increasingly urbanized health landscapes [8]. My work offers a grounded analysis of the gaps and opportunities within the urban ASHA program, emphasizing the need for reforms that are evidence-based but also responsive to the lived realities of CHWs and the communities they serve.

We found that ASHAs experience issues including few formalized recruitment and training processes, insufficient system-level support from dedicated program management staff, and financial burdens. At the core of these challenges are governance-related issues which impact the overall management of Punjab's urban ASHA program to make it more challenging to achieve intended outcomes. Evaluations of the ASHA program in rural settings suggest that greater political and administrative commitment – through institutional structure development, community-led selection processes, robust training, innovative management strategies, and grassroots-level leadership – are essential for a more effective and well-supported ASHA program [2,29]. These lessons are just as – if not more – relevant for urban ASHAs, as urban populations are projected to grow significantly over the next decade [30].

Recommendations to address the identified challenges are summarized below. While the tailored recommendations may address localized challenges, they are grounded in the understanding that improving overall governance is a critical step to enhance the overall effectiveness of the urban ASHA program.

### Recommendation 1: Enhance Community Engagement in Urban Health Programs to Improve ASHA Effectiveness

ASHA guidelines emphasize community participation through involvement in selection, engagement with the MAS, and community group activities [5,31]. Globally, however, community participation in CHW programs is often tokenistic or absent, limiting long-term connections [32,33]. Further, evidence from rural settings in India, where community engagement has historically been stronger, shows that efforts to sustain community engagement structures have tapered off due to unclear objectives, inequitable participation, and weak alignment with local governance structures [34–36]; similar issues exist in urban settings. To overcome this tokenistic engagement, and difficulties with long-term engagement, community engagement should ideally be independently driven, as community engagement that is organized, shaped, and at the discretion of the state often raises questions of robustness and effectiveness [37,38]. In urban settings – comprised

of transient and diverse populations – it may not be practical to select ASHAs in the same way as rural communities do [1]. Districts could implement community-driven selection processes tailored to urban needs by ensuring active participation from the MCs and other local leaders. Urban local bodies, such as MCs, also could benefit from training to effectively engage with ASHAs – ensuring that they are coordinating with and sensitizing the community to the existence and role of the ASHA. Establishing formal collaboration protocols between ASHAs and MCs, including regular interactions and joint planning, could also further strengthen community engagement and align urban ASHA programs with their mandate.

### Recommendation 2: Strengthen Mentorship Structures to Ensure ASHAs Receive Adequate Support

ASHA guidelines specify staffing ratios, including Public Health Managers or Mobilization Officers, and adequate ANM-to-population ratios [31]. However, many of these positions remain vacant, limiting the support available to ASHAs. In the absence of formal training, ANMs have stepped in as informal mentors, leveraging their rapport with ASHAs to provide guidance. To address gaps in the system, the state could consider filling vacant positions or formalizing mentorship roles for ANMs, ensuring they are adequately supported and incentivized. Training ANM workers on aspects of ASHA performance monitoring and supervisory functions, as well as institutionalizing mentorship as a formal component of these roles, could enhance sustainability and build on their existing strong relationships to better support ASHAs.

### Recommendation 3: Implement targeted training for improved system functioning and reduce ASHA workload

Comprehensive training for health workers and community engagement structures may address coordination gaps. ASHAs frequently perform tasks outside their intended scope – often at the direction of their supervisors – highlighting training gaps across all levels. Training senior health workers on vulnerability mapping, as outlined in existing guidelines [5], could help ensure ASHAs focus on populations most in need, reducing their workload. Ongoing, targeted training for ASHAs, senior health workers, MCs, and community structures is critical to improving coordination and service delivery.

### Recommendation 4: Adjust ASHAs' financial compensation to reflect urban realities

Urban ASHAs are often assigned non-health-related duties, often without compensation, detracting from their primary responsibilities. Policymakers could consider establishing clear guidelines to limit these assignments or ensure ASHAs are compensated for such work. States could also revise incentive structures to better reflect urban realities, including accounting for transportation costs through higher base honorariums or transportation allowances. Addressing financial burdens could reduce strain on ASHAs and improve their effectiveness in delivering healthcare services.

The recommendations presented are based on research conducted in two specific urban sites in Punjab. While these findings offer valuable insights into the factors shaping urban ASHAs' experiences and provide guidance for programmatic and policy improvements, they are not universally representative. ASHAs' experiences may vary across districts in Punjab and India due to differences in socioeconomic, cultural, and administrative conditions. However, the challenges identified in this study, such as weak community engagement structures, inconsistent training, and inadequate financial support, align with and address broader gaps observed in the global literature on urban CHW programs [9]. Thus, while this study is context-specific, the findings may have broader relevance for urban CHW initiatives facing similar structural constraints.

The study was intentionally structured as a snapshot of the urban ASHA experience rather than a long-term analysis of ASHA roles, community engagement, or health system dynamics. Further, the ethnographic approach and the inclusion of a finance participatory session were designed to provide more nuanced insights into urban ASHAs' experiences. However, we note that the financial session cannot be taken as conclusive or generalizable, given that the data were generated by ASHAs and not triangulated with bank statements, receipts, or external assessment of time spent per task. This workshop nonetheless highlights the lived experiences and perceptions of ASHAs, showcasing that these ASHAs spend more out-of-pocket than they receive in incentives for most tasks – without accounting for time spent. Moreover, the study's focus

was on the urban ASHA experience in specific sites, acknowledging that variability in ASHA roles across other contexts may present additional complexities beyond the scope of this research.

## Acknowledgments

We are deeply grateful to the ASHAs who generously welcomed us into their lives and work. Their strength, honesty, and commitment to their communities shaped the heart of this research. Through long days, candid conversations, and countless acts of care, they offered us an intimate view of a system too often overlooked. We are honored by their trust and inspired by their perseverance. We also extend our sincere thanks to the Auxiliary Nurse Midwives, Lady Health Visitors, Medical Officers, and health administrators at the district and state levels who shared their time, insights, and experiences with us. Their support and candor helped us understand the broader system in which ASHAs operate. We are grateful to the Punjab National Health Mission and local health authorities for their support in facilitating this work. We also thank the ASHA supervisors and ANM mentors who not only guide ASHAs but also generously supported us in navigating the field. We would also like to thank the following people who generously read and provided inputs on this article: Drs. Nalini Visvanathan, Jill Owczarzak, and Beth Resnick. Finally, we acknowledge the collective labor behind this research – of health workers, communities, and our research team. This paper is a reflection of that shared effort and of our collective hope to strengthen and better support community health systems in India.

## Author contributions

**Conceptualization:** Baldeep Kaur Dhaliwal.

**Data curation:** Baldeep Kaur Dhaliwal, Anuradha Nadda, Shalini Singh, Svea Closser.

**Formal analysis:** Baldeep Kaur Dhaliwal, Madhu Gupta, Anuradha Nadda, Shalini Singh, Svea Closser.

**Funding acquisition:** Baldeep Kaur Dhaliwal.

**Investigation:** Baldeep Kaur Dhaliwal, Madhu Gupta, Anuradha Nadda, Shalini Singh, Aakanksha Dutta, Svea Closser.

**Methodology:** Baldeep Kaur Dhaliwal, Madhu Gupta, Anuradha Nadda, Shalini Singh, Svea Closser.

**Project administration:** Baldeep Kaur Dhaliwal, Madhu Gupta, Shalini Singh.

**Supervision:** Madhu Gupta, Anuradha Nadda, Shalini Singh, Anita Shet, Svea Closser.

**Validation:** Madhu Gupta, Anuradha Nadda, Shalini Singh, Anita Shet, Kerry Scott, Svea Closser.

**Visualization:** Shalini Singh.

**Writing – original draft:** Baldeep Kaur Dhaliwal.

**Writing – review & editing:** Madhu Gupta, Anuradha Nadda, Shalini Singh, Anita Shet, Kerry Scott, Aakanksha Dutta, Svea Closser.

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
