## [Decision Letter · Decision Letter 0]

24 Jun 2025

PGPH-D-25-00991

Urban Community Health Workers in Punjab, India: A Qualitative Study of ASHAs’ Roles in the Health System

Dear Dr. Baldeep Kaur Dhaliwal,

Thank you for submitting your manuscript to PLOS Global Public Health. After careful consideration, we feel that it has merit but does not fully meet PLOS Global Public Health’s publication criteria as it currently stands. Therefore, we invite you to submit a revised version of the manuscript that addresses the points raised during the review process.

We look forward to receiving your revised manuscript.

Kind regards,

Vishal Goyal

Academic Editor

Journal Requirements:

Reviewers' comments:

**Comments to the Author**

1. Have the authors made all data underlying the findings in their manuscript fully available (please refer to the Data Availability Statement at the start of the manuscript PDF file)?

Reviewer #1: No

2. Review Comments to the Author

Reviewer #1: The study addresses a significant gap in the literature by focusing on urban ASHAs, a relatively underexplored area compared to rural CHWs. The study is methodologically sound, well-written, and policy-relevant.

I have a few minor suggestions to further strengthen the manuscript

General comment:

While the study provides valuable insights, its findings are based on two specific urban sites in Punjab. Although the authors briefly acknowledge this limitation, the discussion could be enriched by addressing the potential transferability of findings to other urban contexts. This could be supported by drawing comparisons with existing global literature on urban CHWs.

Specific comment:

• Line 72-77- This section reads more like an interpretation of the findings and would be more appropriately placed in the Discussion section. Consider relocating it there.

• The manuscript would benefit from a brief overview of the Urban ASHA program, including: The total number of Urban ASHAs in India, Key differences between Urban and Rural ASHAs in terms of roles and responsibilities, A short description of the urban health administrative structure, particularly in NUHM settings.

• The manuscript refers to the study locations only as “urban” and “peri-urban” sites in Punjab. If participants anonymity concerns are not the reason for this omission, consider explicitly naming the sites to improve contextual clarity and replicability.

• Line-124-125 stated that “Data not considered as research by participants was excluded from field notes”. It would be helpful to report in the results section how many participants (if any) explicitly declined consent for their data to be used for research purposes

• Line 162 reference to “policymakers and health program administrators”. Consider providing more detail about their official designations or institutional affiliations, as this will help readers understand their influence and relevance.

• Line 196-199 stated that “I began thematic analysis with a close reading… applied it to all the transcripts using MAXQDA…” How were themes developed? Were they inductive (e.g., grounded theory-inspired), deductive (based on literature or theory), or hybrid?

Overall, this is a timely and valuable study that contributes meaningfully to the limited body of knowledge on urban CHWs in India.

Reviewer #2: The topic of this paper is current and of public health importance. Addressing women health and women empowerment.

The narrative style of the case studies has a good flow making it easy to read. The qualitative method is described in details. I also appreciate the reference to the "conceptual depth criteria" approach to validate the method used.

Recommendations are on point, underlying the importance of community engagement and financial support to ensure sustainability of the intervention

---

## [Decision Letter · Decision Letter 1]

29 Jul 2025

Urban Community Health Workers in Punjab, India: A Qualitative Study of ASHAs’ Roles in the Health System

PGPH-D-25-00991R1

Dear Ms. Baldeep Kaur Dhaliwal,

We are pleased to inform you that your manuscript 'Urban Community Health Workers in Punjab, India: A Qualitative Study of ASHAs’ Roles in the Health System' has been provisionally accepted for publication in PLOS Global Public Health.

Best regards,

Dr. Vishal Goyal

Academic Editor
